# Vision and RTLS Safety Implementation in an Experimental Human—Robot Collaboration Scenario

**DOI:** 10.3390/s21072419

**Published:** 2021-04-01

**Authors:** Juraj Slovák, Markus Melicher, Matej Šimovec, Ján Vachálek

**Affiliations:** Department of Applied Informatics, Automation and Mechatronics, Faculty of Mechanical Engineering, Slovak University of Technology in Bratislava, 81231 Bratislava, Slovakia; markus.melicher@stuba.sk (M.M.); matej.simovec@stuba.sk (M.Š.); jan.vachalek@stuba.sk (J.V.)

**Keywords:** human–robot collaboration, safety, RTLS, depth camera

## Abstract

Human–robot collaboration is becoming ever more widespread in industry because of its adaptability. Conventional safety elements are used when converting a workplace into a collaborative one, although new technologies are becoming more widespread. This work proposes a safe robotic workplace that can adapt its operation and speed depending on the surrounding stimuli. The benefit lies in its use of promising technologies that combine safety and collaboration. Using a depth camera operating on the passive stereo principle, safety zones are created around the robotic workplace, while objects moving around the workplace are identified, including their distance from the robotic system. Passive stereo employs two colour streams that enable distance computation based on pixel shift. The colour stream is also used in the human identification process. Human identification is achieved using the Histogram of Oriented Gradients, pre-learned precisely for this purpose. The workplace also features autonomous trolleys for material supply. Unequivocal trolley identification is achieved using a real-time location system through tags placed on each trolley. The robotic workplace’s speed and the halting of its work depend on the positions of objects within safety zones. The entry of a trolley with an exception to a safety zone does not affect the workplace speed. This work simulates individual scenarios that may occur at a robotic workplace with an emphasis on compliance with safety measures. The novelty lies in the integration of a real-time location system into a vision-based safety system, which are not new technologies by themselves, but their interconnection to achieve exception handling in order to reduce downtimes in the collaborative robotic system is innovative.

## 1. Introduction

According to [1], in the majority of cases, the deployment of industrial robots is intended to replace the role of humans in carrying out cyclical, dirty, and dangerous activities. The aforementioned tasks are ideally performed in closed robotic cells. Such cells prevent the accidental entry of humans or other objects into the workplace. The philosophy of the design and principles of robotic cells is proposed in [2]. Problems occur when it is impossible to create a closed cell, or when interaction between a robot and a human is essential. According to [3], the current trends in making collaboration safe consist of pressure mats, light barriers, laser scanners, and the use of collaborative robots. A comparison between classic robots and collaborative robots is provided in [4], showing that the disadvantages of collaborative robots include their lower working speed and higher price. Hence, it is increasingly worthwhile to use external security elements to ensure the safety of the workplace.

An interesting approach to Human–Robot collaboration (HRC) is a contact-type security system called AIRSKIN by Blue Danube Robotics that is described in [5]; the system is a wearable safety cover for robotic manipulators consisting of pressure-sensitive pads. These pads detect collisions with the use of air as the sensing medium. The drawback of this technology is the maximal speed limit of operation and the need to sense objects or persons after contact with the pressure-sensitive pads.

As mentioned in [6], a requirement for implementing HRC is to create a supervised secure and co-operative production envelope between the robot system and a person in the workspace through a virtual protective room referred to as SafetyEYE by the company PILZ. The SafetyEYE system can detect object intrusions into adjustable zones, and based on the type of zone intruded into it responds accordingly, such as by modifying the operational speed of the robotic system. This safety solution has also its drawbacks. It does not provide the possibility to grant security exceptions for expected intrusions within its predefined zones. Another drawback is its lack of human recognition algorithms.

Another current state-of-art safety system mentioned in [7] is from Veo Robotics and called FreeMove. This system monitors work cells in 3D and implements dynamic speed and separation monitoring. This safety system, based on [8], is still awaiting certification. It promises a real-time understanding of the workspace with the possibility to exclude large objects, such as workpieces, so that they do not trigger the safety system. Its disadvantage lies in the versions of robotic controllers it supports, therefore it is suitable only for newer robotic systems.

Such external security elements create zones in the vicinity of the robotic workplace. If something enters such zones, work is slowed or completely halted, as described in more detail in [9]. Security elements based on infrared/light detection only detect their surroundings in one plane. This knowledge has resulted in the idea of monitoring workplaces not only in a single plane but in 3D.

Methods for obtaining 3D data, where every object’s distance from the point of observation is known, have been understood for decades. The most common method is to use two cameras; the basic principles of geometry are dealt with in more detail in [10,11]. Another approach for obtaining 3D data is to use a monocular camera and estimate the distance; this approach is described in more detail in [12,13,14,15]. Other methods include the use of depth cameras. The depth camera principle of operation may be based on the duration of travel of a beam of light (this is explained in more detail in [16,17]), or using structured light that deforms [18]. Current problems that can occur when using camera systems in robotics primarily concern the speed of the processing and evaluation of the 3D data and measurement noise. Despite the current hardware deficiencies, camera image depth processing has been used in interactions between humans and machines for several years. In [19], a stereo camera is used for the teleoperation of a robotic manipulator. Camera systems in co-operation with robots also have applications in medicine, where, according to [20], a camera has been used to calibrate a robot arm during minimally invasive surgery.

The available safety features for collaborating and zoning in the workplace are intended to identify any movement in and intrusion into the work zone. Papers [21,22] address the identification of human beings based on the image processing of human faces or using only images of the ears. Another work [23] addresses the identification of moving objects together with an estimation of their distance from the camera by utilizing one camera. Such techniques make it possible to assign different authorizations depending on the object identified. However, if the objects are externally identical, it is necessary to use additional elements for unambiguous identification. One option is to use Real-Time Location Systems (RTLS) technology, which assigns each object a unique identification number. The primary function of RTLS technology is to track the position and trajectory of objects, as mentioned in [24]. This technology can also be used to optimize trajectories, as stated in [25,26].

This article aims to verify the applications of depth cameras and RTLS to make robotic workplaces safe with zoning. The benefit of this method over the above-mentioned state-of-art technologies lies in incorporating RTLS technology, which will allow selected moving objects to pass through the workplace safety zones without limiting the speed of production. During the work process, moving objects are evaluated, primarily concerning whether or not the object is human. In such a case, for safety reasons no exemption can be granted for entry/transit through work zones without the production speed being restricted.

The novelty consists in the synergy of the interconnection of two co-existing technologies intended in principle for different deployments to increase the safety of robotic workplaces, as mentioned in the article. This design was practically implemented for an experimental robotic workplace, and our results and evaluation are presented below.

This paper is divided into four parts. The first one presents a short theoretical background. The second one contains the main idea of the work and the methodology used. The third part discusses the results. Lastly, the fourth concludes this paper.

## 2. Preliminaries

Quality, accuracy, reliability, and error rates are characteristics that must be evaluated in safety components. The principle of the operation itself often prevents a device from becoming a safety element. Technologies that at one time did not meet safety requirements may, thanks to technological progress, be included in safety systems several years later. In this section, the necessary theoretical knowledge required to understand the subsequent practical implementation is discussed. First, the theory of the safety of robotic workplaces is introduced, followed by knowledge about RTLS and vision technologies.

### 2.1. Safety

Industrial robots are generally large, heavy machines whose movement can be unpredictable. For this reason, it is essential to perform a risk assessment and ensure safety in every robotic workplace. The basic standards addressing the safety of robotic workplaces are EN ISO 10218-1 [27] and EN ISO 10218-2 [28]. These standards define the risks that may be encountered when working with robots and also suitable methods to eliminate them. When deploying a robotic manipulator, it must be ensured that it does not pose a threat to its surroundings in the event of any type of failure (electric, hydraulic, vacuum, etc.). The robot’s working space may be reduced by restricting axes so that it does not cause damage in the event of a fault. The first axis restriction option is achieved by using mechanical stops on the robot itself. The second option is to use external stimuli, where the response time must be taken into account. In situations where physical contact with a robot can occur, it is necessary to set safe distances in the workplace. Safe distances for upper and lower limb reach into the workplace are addressed in EN ISO 13857 [29]. Other standards addressing the safety of robotic systems are ISO 13849-1 [30], ISO 13855 [31], and ISO 12100 [32].

The primary role of the mentioned standards is to prevent contact between humans and robots. However, situations are becoming more common in which, on the contrary, cooperation between robots and humans is expected or even desirable. Hence, the International Organization for Standardization published specification ISO/TS 15066 [33] addressing the safety of collaborative industrial robots. Specification ISO/TS 15066 is a supplement to standard ISO 10218. ISO/TS 15066 proposes power and speed limits when robots and humans collaborate. A risk assessment must also consider where on the body of the operator contact with the robot is most likely to occur. A model of the human body containing 29 specific points categorized into 12 areas was prepared for this reason. Actual contact with a robot is split into quasi-static and transient. During transient contact, some energy is absorbed by the human body. Hence, when a part of the body is not rigidly fixed, it can be subjected to greater forces without serious injury than under quasi-static contact. This applies to the whole body except the head. Table 1 shows speed limits according to ISO/TS 15066 depending on the effective weight of the robot. This value is converted to 1 cm${}^{2}$ of the human body.

According to the international standards ISO 10218-1 and ISO 10218-2, there are four types of collaborative robot use. The first is defined as safety monitored stop. This collaboration method is used when a robot works mostly independently, yet sometimes it is necessary for a human to enter the robot’s workplace. This means that a human must perform specific operations on a product when the product is in the robot’s workplace. The robot must immediately stop all movement as soon as a human enters its workplace. In this case, the robot does not switch off—only brakes are activated. The less time the human needs to spend in the robot’s workplace, the more effective the process is. Hand guiding is another application. This is a type of collaboration used to set a robot’s route by a human guiding the robot’s arm. This is mainly used in cases when it is necessary to teach a robot quick tracks for pick and place operations. The robot must be equipped with an additional device that “feels” the force exerted by the worker on the robot’s tool. Most commonly, this device is a force torque sensor. Collaboration is only possible in learning mode, and additional safety elements are needed in other modes. The second-last type of collaboration between a human and a robot is speed and separation. Here, the workplace is monitored using a laser or camera system. Safety zones are created in the workplace, and the robot’s speed is adjusted depending on the zone in which the human is present. The robot stops when a human enters a danger zone and must wait until this safety parameter is switched off. Individual safety zones can be scaled depending on the distance from the robot. The last variant is power and force limiting. The robot is equipped with sensors in each joint that can detect an external force acting on the joint itself. If a greater than expected force acts on a joint, the robot immediately stops or reverses the movement that led to the initial contact. Such robots are designed to be more sensitive than industrial ones; they are lighter, and their geometry is rounder. A summary of the properties of collaborative industrial robotic systems is presented in Table 2, where the last column shows whether the collaborative type can be adapted into a traditional robotic system.

When choosing an industrial robot, the price needs to be considered in addition to the type of task. Collaborative robots are more expensive, slower, and have worse repeatability than classic industrial robots. As mentioned in [34,35], the price increase of average collaborative robots can be from 50% to over 100%, and is higher compared to non-collaborative ones. It is, therefore, more advantageous to keep the classic industrial robot but supplement it with safety and collaborative elements. In a workplace where human interaction with a robot is expected, the most appropriate collaboration methods are safety monitored stop or speed and separation monitoring. In the first of these, the robot must immediately stop when a human enters its workplace. At higher speeds, a robot takes longer to stop, especially if it is large. As the working speed falls, the time to stop is also dramatically reduced. When locating sensors that detect objects entering the workplace, multiple factors must be taken into account. These factors influence the size of the robot’s working zone. In the case of speed and separation, the robot gradually comes to a stop while its surroundings are continuously monitored. If a human approaches the robot’s working zone, the robot begins to move more slowly and can therefore come to a complete halt more quickly. This is an advantage in silicoations where objects or humans may only be passing by the robotic workplace and a reduction in working speed is sufficient.

The calculation of the minimum size of the danger zone incorporates several factors. The first of these is the speed of the average human *K*, which, according to EN ISO 13,855, is 1.6 m/s. Another factor is the total time to stop *T*. The robot’s reach $S_{R}$, the length of the used tool $S_{T}$, and the length of the working part $S_{W}$ also play essential roles. The maximum values that can occur for $S_{R}$, $S_{T}$, and $S_{W}$ must always considered in the calculations. For laser scanners, the last factor is the supplement *C* (calculated as follows), which prevents reachover:(1)C=1200−(0.4×H),
where *H* is the height of the laser scanner location and 1200 is a distance in mm. The following formula is used to calculate the minimum distance:(2)$$S=K\times T+C+S_{R}+S_{T}+S_{W}.$$

The total time to stop *T* is the sum of the response times $T_{i}$ of each device *i*. If a scanner is used as a safety element, it is necessary to take into account the response time of the laser scanner $T_{S}$, the robot $T_{R}$, and the robotic controller $T_{C}$, as well as the time to perform the logic cycle $T_{LogicCycleTime}$. If additional equipment such as a Programmable Logic Controller (PLC) is used in the safety system, the response time for each must be calculated separately. The resulting relationship for determining the minimum size of the danger zone then looks as follows:(3)$$S=K\times \left(T_{1}+T_{2}+...+T_{n}\right)+C+S_{R}+S_{T}+S_{W}.$$

Parameters such as $T_{LogicCycleTime}$, $T_{R}$, *C*, $S_{R}$, and $S_{T}$ must be provided by the manufacturer. The size of the warning zone or other safety zones is not subject to such strict conditions as for determining the minimum size of the danger zone. An example of minimum distance calculation is shown in Figure 1, where (1) is the danger zone, (2) is the warning zone, and lastly (3) is the safe zone.

### 2.2. Real-Time Location Systems

RTLS have multiple uses. The first is the identification of a tag and its carrier, while others include the location of the carrier and the tracking of its movement in real time. This technology is primarily intended for the indoor positioning and tracking of objects, with an accuracy of up to tens of centimetres in the case of Ultra-Wide Band (UWB) technology.

As mentioned in [36], tracking the position of tags in RTLS systems is possible using several methods, mostly based on determining position using a function of distance travelled from the time needed to overcome this dependence. A method that works only on this principle is called Time of Arrival (ToA). A more accurate method that enables smoother measurement is the Time Difference of Arrival (TDoA) method, which uses time differences for signals received from neighbouring transmitters, not only absolute values. This implies the need to synchronize the times of the location system devices (transmitters and receivers). Another method we could mention is Two-Way Ranging (TWR). Communication begins with an anchor sending a message, after which a tag receives the message and responds to the message after a time delay. After the response is received, it is possible to calculate distance based on the message transmission speed and the total time between sending the message and receiving the response.

In this paper, the location system uses the previously mentioned UWB technology, which is based on low-power ultra-wide wireless communication. Signals are sent in short pulses (0.16 ns) in a wide spectrum (500 MHz) on multiple frequency channels at once. According to [37], UWB does not interfere with other transmissions in the same frequency band. In addition, it eliminates the risk of capturing the reflection of the sent signal from surrounding objects. With such fine time sampling, it is possible to capture and distinguish such signal reflections, making the technology more accurate and robust.

When constructing an RTLS system based on UWB technology, it is important to maintain a clear line of sight, meaning that all system devices can see each other. Every obstacle reflects and absorbs parts of the signal, depending on its radio transparency and material type. This causes positioning inaccuracies and reduces the system range. Figure 2 shows how the UWB transmitter sends short signal impulses in a wide range, and the pulse occupies a wide frequency band.

### 2.3. Vision-Based Detection Technology

Similarly to human eyes, vision systems provide a massive amount of information about the surrounding environment. According to [38], the current level of vision systems is comparable to human vision. In addition, vision systems can calculate the distance of each pixel and so we are able to create a depth map of the surroundings. A single depth camera can replace several sensors by utilizing its knowledge of objects’ spatial positions and their identification.

To date, most proposed vision-based recognition methods, as mentioned in [39], have used additive Red Green Blue (RGB) colour model-based images for object detection. Nonetheless, as presented by [40], contrast, entropy, correlation, energy, the mean, and the standard deviation can be calculated from examined images. As stated in [41], the use of the RGB colour model images has its advantages compared to grayscale ones by providing three times more data, but it still has significant drawbacks. It omits the scene’s depth information when converting from three into two dimensional space and providing the mentioned illumination, texture, and colour information. This is why object recognition methods based on RGB may be affected by external factors, which could significantly hinder their usage in most practical applications.

Depth cameras are the result of progress in research and the decreasing costs of depth sensors integrated with RGB cameras. This has led to the implementation of Red Green Blue-Depth (RGB-D) images, which define the RGB values and the distance between the RGB-D sensor of the camera and the captured object of interest. As mentioned in [42,43], depth information is obtained by stereovision [44], time of flight (TOF) [17,45], or structured light [46] depth cameras. The described types of depth cameras are shown in Figure 3. According to [42,43], the working principles of depth cameras can be shortly described as follows:Stereovision cameras use the triangulation principle for the calculation of the depth of a captured scene. The use of two or more image sensors or cameras is needed to perceive depth, simulating human binocular perception.TOF cameras emit infrared (IR) light, which is reflected back to the camera sensor by objects in the captured scene.The last working principle is cameras based on capturing structured light. The camera consists of an IR projector and a capturing sensor, which has similarities to stereovision cameras, with the difference of replacing the second image sensor with a projector.

## 3. Problem Statement and Main Work

Following the introduction, this paper aims to propose an experimental vision and RTLS-based HRC setup that should provide an approach other than the reviewed state-of-art technologies. Our aim was to analyze the downsides and try to improve them. Other states-of-art technologies provide multiple solutions to ignore or exempt areas from the workplace that are not monitored by the security system. This is beneficial, but it also has drawbacks. Excluding certain spots within the workspace from virtual fencing could create security risks. This paper’s approach addresses the issue of dynamically recognizing objects which could come from any direction with a depth camera. The solution also extracts the image features and checks with a certain confidence level if the recognized object is a person. If the recognized object is a person, no exceptions are granted, and the system reacts like any speed and separation monitored (SSM) HRC system, where the distance from the person to the robotic system is cyclically compared. If the person enters warning zones, the system slows down the robotic system’s speed, and if the intrusion continues into the danger zone it completely stops the robotic system. The recognized object is also checked against the RTLS tag position data to check if the tag ID has granted exceptions. If such a case exists, the security system ignores these intrusions only if the object is not detected as a person and allows it to move around the workplace without modifying the speed of the robotic system. The results of this experimental collaborative robotic workplace are presented in the corresponding section, which is followed by a discussion and conclusion.

### 3.1. Test Workplace Specifications

The HRC safeguarding of the operator on the test workplace is achieved through SSM by maintaining a certain minimum separation distance during operation. The intelligent zoning design was implemented in a functioning robotic workplace. The function of this workplace is the bin picking of objects that are then assembled using a robotic manipulator. The maximum reach of the manipulator is 540 mm; its maximum speed in T1 mode is 250 mm/s, and in T2 mode it should achieve a theoretical maximum of up to 2 m/s, based on [47,48]. A 3D scanner located above the robotic workplace scans a bin containing objects. After identification, the scanner sends the object’s coordinates to the robot control system, which sends the trajectory information to the robotic manipulator. The manipulator grabs the object and assembles it with another object. The supply part of the process is carried out using an autonomous supply trolley. The workplace floor plan is shown in Figure 4. As the workplace area is limited, it was necessary to modify the individual zones’ size to perform experiments for various depth camera settings.

The primary workplace components used for this article’s preparation were a KUKA KR3 R540 robotic manipulator (the main specifications can be found in the datasheet [47,48]), a Stereolabs ZED depth camera (the datasheet can be found in [49]), a Pozyx UWB real-time location system (details can be found in [50]), a Photoneo PhoXi 3D scanner (details can be found in [51]), an internally developed supply trolley, and a computer. The entire process can be split into two parts. The first is robot control, consisting of the bin picking of objects performed by the robotic manipulator and an automated supply by the mobile trolley. The second part is the supervisory control that monitors and zones the robotic manipulator’s working environment and, depending on the external stimuli, adjusts the performed task’s speed. Safety zones are then created using the input data from the depth camera. The UWB-based location system operates concurrently to monitor the spatial position of the tags. The depth camera is directly connected to the computer. In contrast, the RTLS system is first connected to an embedded platform that uses the OPC Unified Architecture (UA) protocol to communicate with the computer. The computer then evaluates the input data and sends the information using the TCP/IP protocol to the robot control system. The data received by the robot control system are processed by the KUKA user interface KSS (KUKA.SystemSoftwarere), more precisely by a submit file (SF). The SF is a background process inside the user interface, responding cyclically to changes using its preprogrammed logic. In other words, the SF is superior to the user interface. In this case, the SF sets the speed or halts the robot motors based on the computer’s input data. The KSS user interface contains functions for the trajectory planning of the robotic manipulator, and these functions are slaves to the SF settings. A simplified architecture and the method of communication between the components of the security workplace are shown in Figure 5.

RGB-D workplace image processing is performed by a ZED depth camera, where the area covered by the depth camera is presented in Figure 4. The depth camera itself can operate in four resolution modes, ranging from the lowest-quality Wide Video Graphics Array (WVGA) at a resolution of Side by Side 2 × (672 × 376) pixels through to a 2.2 K output at a resolution of 2 × (2208 × 1242) pixels. Depending on the mode, it is possible to set the frames per second between 15 and 100 Hz. Depth recording uses the same output resolutions as the RGB recording. The depth range of the camera is between 0.2 and 20 m. The camera’s field of view (FOV) is 110° × 70° × 120° (horizontal × vertical × diagonal). An 8-element wide-angle all-glass lens with optically corrected distortion and an f/1.8 aperture is used. A USB 3.0 port is used for both connection and power supply. The camera is compatible with the Windows 10, 8, 7; Ubuntu 18 and 16; and Jetson L4T operating systems. The camera enables integration with third-party SW, including Unity, OpenCV, ROS, Matlab, Python, and C++. The manufacturer’s minimum system requirements are a dual-core 2.3 GHz processor with 4 GB RAM and an NVIDIA graphics card with a compute capability of over 3.0. The camera’s position was optimally chosen so that it was possible to capture the available space around the robotic workplace.

The Pozyx location system based on UWB radio waves forms a separate part of the workplace. This location system requires fixed anchors, tags, and a computing unit to process the data. The placement of the anchors influences the measurement accuracy. Figure 4 shows how the anchors are arranged on the ceiling within the room to illustrate their position accurately as possible. When placing the anchors, the essential advice in [52] was followed to achieve maximum measurement accuracy. The anchors use the TWR positioning principle and need a continuous 3.3 V power supply, as they do not have batteries. The Decawave DW1000 transceiver allows a communication speed of up to 6.8 Mbps. The selected tags are shields compatible with Arduino via an I2C bus. The board also includes an STM32F4 microcontroller. The tags contain the same transceiver as the anchors and also require a continuous power supply. The manufacturer states an accuracy of 10 cm in a noise-free environment, which is sufficient for our purposes. The initial data processing is performed on an embedded Raspberry Pi 3 platform. Several tasks are performed on this embedded platform—e.g., reading data, noise filtering, and then sending the processed data using the OPC UA protocol to the computer. The use of an embedded platform has proven to be a satisfactory solution.

The main part of the algorithm is run on a computer with a 10th-generation Intel Core i5 processor and a quad-core processor running at 2.5 GHz with the option of overclocking to 4.5 GHz. The computer has 16 GB DDR4 RAM running at 2.93 GHz. A 1 TB SSD provides sufficient storage and enables high read and write speeds. A graphics card that supports the Compute Unified Device Architecture (CUDA) platform is an essential part of the computer. In this case, an Nvidia GeForce RTX 2060 with 6 GB RAM has been used. This graphics solution was needed to provide compatibility between the camera system and the computer itself.

### 3.2. Depth Recording Processing

Depth monitoring at a robotic workplace is achieved using a depth camera working on the passive stereo principle. The camera manufacturer offers interfaces for several programming languages; Python was used here. An API was created to read data from the depth camera with the subsequent detection of moving objects. As the camera uses the passive stereo principle, it is possible to access the left and right video streams separately and the depth recording. The depth recording from the camera is stored as a Point Cloud in the variable Point3D. Let the Point Cloud be represented as $Point3D=\{P_{ij}\}$, where $P_{ij}\in R^{4}$ is a layer of information defined as $P_{ij}=\left[x_{ij},y_{ij},z_{ij},RGBA_{ij}\right]$, with i=1,…,W, j=1,…,H.

The user selects the width *W* and height *H* of the stream—see Section 3.1. The Point Cloud is composed of data from four 32-bit channels containing $\left[x_{ij},y_{ij},z_{ij}\right]$ coordinates for the pixel $P_{ij}$ relative to the center of the camera and an $RGBA_{ij}$ component, which contains colour information, where *A* is the alpha channel.

To create the safety zones, the principle whereby an initial reference recording of the room is created to which each measurement is compared was used. The reference recording contains the distance of each pixel to the center of the camera. The initial scan quality is critical, as any error is transferred to all the other measurements. The Python programming language offers the OpenCV library that enables two images to be compared and find the differences. This work does not compare camera recordings based on changes in the colour component but rather changes in depth. Therefore, it is necessary to convert the Point Cloud data into a form used by the OpenCV library by removing the RGBA element using the method delete(Point3D,3,axis=2) contained in the NumPy library. The depth change between the reference and the actual recording is defined as $Point3D_{diff}$. For this subtraction operation, we use the subtract method included in the NumPy library: $subtract(Point3D_{reference},$$Point3D_{actual})$. $Point3D_{diff}=\left\{P_{ij}\right\}$ with the same dimensions as the image, where information about the RGBA component is eliminated and information about the size of the coordinate change in the *X*, *Y*, *Z* axes is retained for each pixel—i.e., $P_{ij}=\left[\Delta x_{ij},\Delta y_{ij},\Delta z_{ij}\right]$. Figure 6 illustrates the output from the depth camera.

The values in $Point3D_{diff}$ can be immediately adjusted and possible measurement-related errors eliminated. The maximum change value is set based on the geometric dimensions of the monitored room. If this value is exceeded, it is clear that an erroneous measurement has occurred for the relevant pixel. Another type of undesirable measured value is a very small number—the result of measurement noise. An experimentally minimum change value for a shift in one axis of 0.3 m was set. At this value, all the undesirable noise that occurred was eliminated. This value was also sufficiently high to enable the confident identification of a human and a mobile supply trolley. The last type of undesirable value that occurs in this part is NaN (not a number). By observation, it was established that these values tended to occur at the edges of the recording. All these types of undesirable values are a negligible part of the depth recording and can be eliminated without adversely affecting the image recognition. The elimination of the undesirable measurement values was performed by replacing them with the value 0. The zero values were acquired by pixels where there was no change. After the adjustment, $Point3D_{diff}$ contains real values ranging from 0.3 m to the room’s maximum distance. As the RGB component of each image contains values from 0 to 255, scaling is required using the following relationship:(4)$$value_{new}=\left(255-0\right)\times \frac{value-value_{min}}{value_{max}-value_{min}}+0,$$
where 0 is the lower value of the new interval, 255 is the upper value of the new interval, and $value_{min}$ and $value_{max}$ are the limit values measured based on the geometry of the room. After rounding to whole numbers and the conversion to grayscale, it is possible to proceed to the threshold. The threshold function compares the pixel value with the threshold_value. If this value is higher than the threshold_value, the pixel value is replaced. This replacement creates a new image consisting of only two colours. The OpenCV library provides several threshold types. After testing various threshold methods, the most suitable choice appears to be Otsu’s binarization. Otsu’s binarization works with bimodal images. These are images whose histogram has two peaks. According to [53], the threshold value is set as the mean value between these two peaks.

As mentioned in [54], Otsu’s algorithm tries to find a threshold_value$\sigma_{w}\left(t\right)$, which minimizes the weighted within-class variance given by the equation:(5)$$\sigma_{w}^{2}\left(t\right)=q_{1}\left(t\right)\sigma_{1}^{2}\left(t\right)+q_{2}\left(t\right)\sigma_{2}^{2}\left(t\right),$$
(6)$$\begin{array}{cccc} q_{1}\left(t\right) & =\sum_{i=1}^{t}P\left(i\right), & q_{2}\left(t\right) & =\sum_{i=t+1}^{I}P\left(i\right), \end{array}$$
(7)$$\begin{array}{cccc} \sigma_{1}^{2}\left(t\right) & =\sum_{i=1}^{t}{\left(i-\mu_{1}\left(t\right)\right)}^{2}\frac{P(i)}{q_{1}\left(t\right)}, & \sigma_{2}^{2}\left(t\right) & =\sum_{i=t+1}^{I}{\left(i-\mu_{2}\left(t\right)\right)}^{2}\frac{P(i)}{q_{2}\left(t\right)}, \end{array}$$
(8)$$\begin{array}{cccc} \mu_{1}\left(t\right) & =\sum_{i=1}^{t}\frac{iP(i)}{q_{1}\left(t\right)}, & \mu_{2}\left(t\right) & =\sum_{i=t+1}^{I}\frac{iP(i)}{q_{2}\left(t\right)}, \end{array}$$
where $q_{1}\left(t\right)$ and $q_{2}\left(t\right)$ are the probabilities separated by a threshold *t*, $\sigma_{1}\left(t\right)$ and $\sigma_{2}\left(t\right)$ are the variances, $\mu_{1}\left(t\right)$ and $\mu_{2}\left(t\right)$ are the mean, P(i) is the probability, *i* is the quantity of pixels, and *I* is the maximum pixel value (255).

The OpenCV library also includes another essential function: findContours. A contour is a curve connecting all the continuous points with the same colour. This function requires three parameters: a grayscale image, a hierarchy between contours, and an approximation method. The contour hierarchy is the relationship between the parent (external) and child (internal) contour. In this work, we selected the parameter Retr_External for the hierarchy. This setting only takes parent contours into account, as it is necessary to identify an object as a whole. The last parameter determines whether the contour should remember all the boundary points. This option was not needed, so the approximation method SIMPLE, which only remembers the two endpoints, was selected. Figure 7 shows the highlighting of an object that entered the workplace after threshold application, and Figure 8 shows the same identified object on the depth recording of the robotic workplace.

Once an object is identified, it is possible to proceed to the computation of each pixel’s distance to the robotic workplace and establish the shortest distance. It is necessary to find the shortest distance for each object as a whole. Each created contour represents one object. The algorithm cyclically scans all the values of the pixels in the contour and calculates the distance of the given pixel *D* to the center of the robotic workplace using the formula:(9)$$d={\left(x_{0}-x_{1}\right)}^{2}+{\left(y_{0}-y_{1}\right)}^{2}+{\left(z_{0}-z_{1}\right)}^{2},$$
(10)$$D=\sqrt{d},\mathrm{if}d>0,$$
where $x_{0}$, $y_{0}$, and $z_{0}$ represent the displacement for each pixel read from Point3D and $x_{1}$, $y_{1}$, and $z_{1}$ are the distances between the center of the camera and the center of the robotic workplace. The camera is 2.5 m above the ground and 0.9 m behind the robotic workplace, and, in addition, is rotated at an angle of 45°. The camera axis $x_{cam}$ is the same as that of the robot axis $x_{rob}$ and thus $x_{1}=0$. The values $y_{1}$ and $z_{1}$ are subsequently computed using Formulaes (Equation 11) and (Equation 12), which stem from Figure 9. Figure 9 shows the relative positions of the robot and camera coordinate systems.
(11)$$\sin\alpha =\frac{a^{\prime }}{b^{\prime }}\to a^{\prime }=\sin\alpha \times b^{\prime },a^{\prime }=y_{1},$$
(12)$$\frac{b}{b^{\prime }}=\frac{c}{c^{\prime \prime }}\to c^{\prime \prime }=\frac{c\times b^{\prime }}{b},c^{\prime \prime }=z_{1}.$$

### 3.3. Human Recognition

After successfully processing the depth image and identifying the objects moving in the robotic workplace, it is vital to establish whether a given object is a human. It is not suitable to apply human recognition to the depth recording. As a stereo depth camera is used, left and right RGB streams are available. The data from only one of the streams are sufficient. The data contain the RGB component of each pixel. The actual identification of humans is performed using the Histogram of Oriented Gradients (HOG). HOG is a feature descriptor that focuses on the structure or shape of an object. Unlike edge features, which can only establish whether a given pixel is an edge, HOG is capable of establishing both an edge and its direction. This is achieved by dividing the image into small regions and a gradient, and the orientation is computed independently for each region. Finally, a histogram is created for each region separately. More information about the functioning of HOG and its application in identifying humans can be found in the work [55].

The HOG algorithm is part of the OpenCV library. When initializing HOG, it is possible to select a preprepared human detector that has already been trained to identify humans. The used detectMultiScale function for detecting humans has four input parameters: image, winStride, padding, and scale. The winStride parameter defines the step size of the detection window in the horizontal and vertical directions. Smaller parameter values lead to a more accurate result, but also extend the computational time. The padding parameter indicates the number of pixels in the *x* and *y* directions of the detection window. The last parameter, scale, determines the factor according to which the window is resized in each layer of an image pyramid. Again, the smaller the scale parameter value, the more accurate the result, but the longer the time needed for the computation. The detectMultiScale function outputs two attributes. The first attribute rects contains information about all the boundary boxes. This information is in the format of the coordinates of the lower left point and the width and height of the individual box. The second attribute is weight, containing the confidence scores.

The human recognition subroutine is part of the algorithm for identifying objects from the depth recording. Every object that enters the workplace is identified through the change in depth and marked with a contour. The contour is defined using the coordinates of the left lower point and the width and height. The identification of humans alone is not sufficient to verify whether there really is a person in the workplace. A better option is when the same pixels, representing the depth change and human identification, overlap. It is possible to verify overlap using the conditions $L_{1x}\geq R_{2x}$ or $L_{2x}\geq R_{1x}$ and $L_{1y}\geq R_{1y}$ or $L{}_{2}y\geq R_{1y}$, where $L_{1}$ and $R_{1}$ are the coordinates of the lower left and upper right point of the box, indicating a potential person in the image. Similarly, $L_{2}$ and $R_{2}$ are the coordinates of the contour bounding the depth change. If none of the conditions apply, there has been an overlap. Figure 10 and Figure 11 show the identification of an object from the depth recording and the verification of a person’s presence in the same recording.

### 3.4. Supply Trolley Recognition Using RTLS

Autonomous supply trolleys are nothing new in the industry. There are usually several such trolleys in a factory, and they all generally look alike. Although a camera can identify them in space, it cannot determine which of the trolleys it sees. The trolleys need to be equipped with unique ID numbers. In such cases, it is possible to apply a location system based on the UWB principle.

The RTLS sends information via an embedded platform about each tag’s ID and location to a computer. The RTLS works with its own coordinate system, and so the coordinate systems need to be unified. A procedure was chosen whereby the camera coordinate system is converted to the RTLS coordinate system. This conversion is performed only for the point with the smallest value for each identified object. The point with the smallest value represents the shortest distance to the robotic workplace. The location data are read from Point3D. Using basic trigonometric functions and the robot’s known location in the RTLS coordinate system, the coordinates of the nearest point are subsequently converted to the RTLS coordinate system. It is crucial to consider several factors during the actual comparison of whether the location from the RTLS is the same as the location from the camera, as shown in Figure 12. Both systems work with a certain degree of inaccuracy. The inaccuracy of the camera increases with distance, while the RTLS inaccuracy depends on several factors. The inaccuracy is increased in a working environment where wave reflections occur. Another possible factor could be the location of the tag on the supply trolley. As in this application, the RTLS is not used for precise positioning but to identify a supply trolley—accuracy is less critical.

## 4. Results and Discussion

During the experiments, the workplace’s primary safety elements—including the safety laser scanner—were switched off. The robotic manipulator worked in automatic mode but at a reduced maximum speed. An authorized person supervised the workplace for the whole period to avoid damage to health or property. Due to a lack of space, the workplace was split into three zones: danger, warning, and safe. However, the safety algorithm is not limited by the number of zones. The dynamic safety system’s functionality using the depth camera and RTLS was verified for two simulation scenarios.

The first scenario was the regular operation of the robotic workplace, including its supply. In this option, people were not allowed to enter the robotic workplace, and hence the part of the safety algorithm used to identify humans was omitted, as is shown in Algorithm 1. Trolleys that do not have a tag can enter the workplace and are not detected by RTLS, so using only RTLS to determine the objects in the workplace is not a satisfactory safety source. The objective of this scenario was to verify the reduction in the computational time needed for the safety algorithm. The processing and evaluation of the data from the depth camera were performed on a computer. The computation time needed for the safety algorithm differs depending on the number of objects for which it is necessary to calculate the smallest distance to the robotic workplace. In this scenario, the measurement was carried out separately in a case where supply trolleys were carrying out their tasks at the workplace retained. The second case was where no supply trolleys were at the workplace and the only thing moving was the robotic manipulator. The actual movement of the robotic manipulator is evaluated as movement around the workplace. Therefore, the area in which the manipulator performs its tasks is ignored, resulting in a further reduction in the computation time. The resolution of the depth camera also influences the speed of the safety algorithm. Figure 13 and Figure 14 show the time it took for the safety algorithm to process a single image for different depth camera settings. It is also possible to see the percentage deviation between the actual measurement and the reference recording.

The scenario’s objective was to grant an exception for a supply trolley whose movement trajectory was known. The supply trolley had a unique ID and did not supply the given workplace—it only passed through the safety zones. The given trolley did not affect the operation or safety of the robotic workplace. According to the workplace’s safety elements, the manipulator speed should be adapted as soon as the trolley entered the safety zone. The granting of the exception meant that even if the trolley passed through the safety zone, the robotic workplace’s speed did not change. Of course, this applies only if no other object enters the zone. A trolley supplying the given workplace directly interferes in the production process and does not have an exception at this workplace, so the speed changed in this case. Many factors are taken into account when evaluating exceptions, including the size of the material supplied, the distance between the supply route and the robotic system, and even the possibility of collision. Hence, a safety technician must be responsible for granting exceptions in practice. The presented safety system results in an increase in the rate of work without the need for downtime.

The second simulation scenario anticipates the presence of a human at the workplace in addition to supply trolleys. The primary part of the safety algorithm is the same for both scenarios. In this case, there is also an incorporated subroutine for recognizing humans, as is represented by Algorithm 2. The speed and quality of human detection are directly dependent on the entry parameters winStride and scale. These parameter values were experimentally selected: winStride=(4,4) and scale=1.175. When setting the parameter values, the emphasis was on ensuring that the camera identified the person in every accessible part of the robotic workplace.On the other hand, these parameter values resulted in an increase in the safety algorithm’s computational time. A longer processing time for each image increases the total response time, which is part of Equation (Equation 3) and thereby increases the size of the zone itself. The security algorithm was tested using two cases, as in the first simulation scenario. In the first case, there were no trolleys or people in the workplace and the robotic manipulator was the only thing that moved. In the second case, supply trolleys moved around the workplace and people randomly passed through the workplace. The experiment was performed with different camera settings. The results in Figure 15 and Figure 16 show the computing time per cycle and, in addition, the deviation parameter between the reference recording and the actual measurement.
**Algorithm 1** Security algorithm without human recognition     **Input:** Stereo stream from the depth camera and tag position from the RTLS system.     **Output:** Robotic system speed (%).1:Initialize depth camera parameters (resolution) and RTLS system (OPC UA).2:Set $x_{R-C}$, $y_{R-C}$, $z_{R-C}$, $x_{RTLS-R}$, $y_{RTLS-R}$, and $z_{RTLS-R}$ as the known dimensions between the depth camera (C), robotic system (R), and beginning of the RTLS coordinate system.3:Save the reference depth image into $point3D_{reference}$.4:**while** the robotic workplace is switched on **do**5:    Save actual depth image into $point3D_{actual}$.6:    Determine $point3D_{diff}=substract\left(point3D_{reference},point3D_{actual}\right)$.7:    Convert $point3D_{diff}$ to grayscale.8:    Eliminate initial measurement errors.9:    Scale the $point3D_{diff}$ values using Equation (Equation 4).10:    Convert into a binary image using the function cv2.threshold.11:    Contour search with the function cv2.findContours.12:    Perform distance calculation between each value in the contour and the robotic     system $dist=\sqrt{{\left(value_{x}-x_{R-C}\right)}^{2}+{\left(value_{y}-y_{R-C}\right)}^{2}+{\left(value_{z}-z_{R-C}\right)}^{2}}$.13:    Convert $min_{dist}$ coordinates to RTLS system coordinates.14:    Find all contours where the RTLS system coordinates match with the contour coordinates and the RTLS tag ID is set to have an exception.15:    **if** no other contours are found **then**16:        Set robot_speed=20.17:    **else**18:        find $min_{dist}$ of the remaining contours.19:        **if**
$min_{dist}$≤danger_zone_size
**then**20:           Set robot_speed=0 for the danger zone.21:        **else if**
warning_zone_size≥$min_{dist}$>danger_zone_size
**then**22:           Set robot_speed=10 for the warning zone.23:        **else**24:           Set robot_speed=20 for the safe zone.25:        **end if**26:    **end if**27:**end while****Algorithm 2** Security algorithm with human recognition     **Input:** Stereo stream from the depth camera and tag position from the RTLS system.     **Output:** Robotic system speed (%).1:Initialize depth camera parameters (resolution) and RTLS system (OPC UA).2:Set $x_{R-C}$, $y_{R-C}$, $z_{R-C}$, $x_{RTLS-R}$, $y_{RTLS-R}$, and $z_{RTLS-R}$ as the known dimensions between the depth camera (C), robotic system (R), and the beginning of the RTLS coordinate system.3:Save the reference depth image into $point3D_{reference}$.4:**while** the robotic workplace is switched on **do**5:    Save actual depth image into $point3D_{actual}$.6:    Save actual image into image.7:    Search for the coordinates of potential human rects in the picture using function    hog.detectMultiScale.8:    Assign coordinates rects to $L_{1}$ and $R_{2}$.9:    Determine $point3D_{diff}=substract\left(point3D_{reference},point3D_{actual}\right)$.10:    Convert $point3D_{diff}$ to grayscale.11:    Eliminate initial measurement errors.12:        Scale the $point3D_{diff}$ values using Equation (Equation 4).13:        Convert into a binary image using the function cv2.threshold.14:        Perform contour search with the function cv2.findContours.15:        Assign contour coordinates to $L_{2}$ and $R_{2}$.16:        Verify that the contour corresponds to the identified person     $doOverlap(L_{1},R_{1},L_{2},R_{2})$.17:        **if**
doOverlap==True
**then**18:           Set humanPresence=True.19:       **end if**20:       Calculate the distance between each value in the contour and the robotic system $dist=\sqrt{{\left(value_{x}-x_{R-C}\right)}^{2}+{\left(value_{y}-y_{R-C}\right)}^{2}+{\left(value_{z}-z_{R-C}\right)}^{2}}$, for dist>0.21:        Find the $min_{dist}$ of each contour.22:        Convert $min_{dist}$ and $min_{human}$ coordinates to RTLS system coordinates.23:        Find all contours where the RTLS system coordinates match with the contour coordinates and the RTLS tag ID is set to have an exception.24:        **if** no other contours are found **and**
humanPresence==False
**then**25:           Set robot_speed=20.     **else**26:           find $min_{dist}$ of the remaining contours.27:           **if**
$min_{dist}$≤danger_zone_size
**then**28:             Set robot_speed=0 for the danger zone.29:           **else if**
warning_zone_size≥$min_{dist}$>danger_zone_size
**then**30:              Set robot_speed=10 for the warning zone.31:           **else**32:              Set robot_speed=20 for the safe zone.33:           **end if**34:        **end if**

The granting of an exception to transit or enter the safety zone worked on the same principle as in the first simulation scenario. In this case, however, it was ensured that the person’s location took priority during the granting of the exception. This change means that an exception cannot be granted for a supply trolley if a person enters a safety zone with a trolley in it. Consider safety zone 1 as the nearest to the robotic system and safety zone 3 as the farthest from it. If a supply trolley with an exception is in safety zone 2 and a person is in safety zone 3, then the trolley will be granted an exception. We have also considered that the person will have a tag on them, and this tag has an exception. In this case, the algorithm will cancel the exception and evaluate the situation in the same way as when a person without a tag enters the workplace. This simulation scenario allocates a higher priority to a person than to other objects.

Both scenarios were tested for the presence of multiple people or trolleys at the same time. The algorithm can identify several objects simultaneously. The more objects there are in the workplace, the more distances between the object and the robotic workplace must be calculated by the algorithm, which is reflected in the speed of the algorithm. However, the safety algorithm is not intended for human–trolley interaction.

A switch to a higher camera resolution causes an increase in measurement or background noise, which can be seen on the graphs on Figure 13, Figure 14, Figure 15 and Figure 16. The Testing workspace does not have adequate protection against adverse lighting effects. The depth camera’s principle is based on passive stereo, therefore the sunlight strongly affects the lightning conditions. This impact is shown in the second graph in Figure 15, where the brightness changes continuously during the measurement, together with external lighting conditions interfering with the robotic workplace. Meanwhile, the graph in Figure 17 shows the effect of outdoor lighting condition changes during the day when comparing the reference image with the actual image. The reference record was made in the morning when the sun was shining into the room. A subsequent comparison of the current record with the reference shows that the sun’s rays stopped entering the room at around 09:30. Between 09:30 and 14:30, the lighting conditions were relatively constant, and sunset began around 14:30. To solve this problem, it is necessary to add a reference image refresh function after some time when no moving objects are detected. This reference refresh could solve the issue of significant light differences over time in workplaces. A more extended algorithm computation time means an increase in the safety system’s total reaction time and thus a necessary enlargement of the safety zones.

The evolution of computational time per cycle with respect to image quality is shown in Figure 18. Based on the approximation of the data obtained from the measurements, it is possible to predict the average calculation time for the depth camera’s 2.2K resolution. The prediction is shown in Figure 18 and should be around 0.66 s without HOG and around 1.89 s with HOG. A higher camera resolution does not allow for the quick detection of a person’s presence within the monitored workspace. However, this deficiency can be compensated for by increasing the safety zone.

Human recognition around the robotic workplace was verified for various scenarios. Figure 19a–c aims to identify a person in different lighting conditions, different positions in the room, and different clothing colours. Meanwhile, Figure 19d–f focus on identifying a person who is not fully visible in the camera stream. We found that the HOG algorithm can just as successfully identify moving people as static ones. Identification problems occurred when the person was directly under the camera, regardless of whether they were moving or not. The reason for this is that, when viewed from above, human proportions are lost from the camera’s view.

The speed and accuracy of the RTLS have a significant impact on the correct identification of the supply trolley. Based on the advice in [56], three measurement points were selected for testing the accuracy. These points reflect the most common cases that can occur. The positions of the measurement points are shown in Figure 4. During measurement, the tag was placed on a static supply trolley. Point A on Figure 4 represents an ideal case when no object is causing interference in the proximity of the tag other than the trolley—clear line of sight. The most common variant in practice is that around the tag there are objects causing interference, such as metals—measurement point B. The last point, C, presents a case where the tag is very close to one of the anchors.

Different UWB communication settings result in different accuracies and total measurement times. Key parameters are the bitrate and preamble length. Parameters such as channel and pulse repetition frequency do not significantly affect the accuracy and measurement speed. Table 3 compares the differences between the different UWB settings. The bitrate of 110 kbit/s and preamble length of 2048 symbols represent slow communication with a large amount of data; the opposite values and behaviors are shown for the bitrate of 6810 kbit/s and the preamble length of 64 symbols. The last option is something between these two settings. The third column of Table 3 shows the measurement accuracy at different settings and different locations. The smaller the dispersion, the more accurate the measurement, because the points are closer to each other. The most suitable parameters are 850 kbit/s for the bitrate and 64 symbols for the preamble length, which were also used during the testing of the security algorithms. The update rate did not fall below 35 Hz for these values. As the supply trolley has a maximum speed of 0.65 m/s, the selected UWB parameters are sufficient for the given application. The trajectory of the trolley movement during the measurement did not affect its identification by RTLS.

## 5. Conclusions and Future Work

The contribution and novelty of this paper can be seen in the practical feasibility of the approach proposed and its improvement on the state-of-the-art technologies, as mentioned in the introduction. Thus, we have aimed to improve and expand these concepts and create a different view of the safety of robotic collaborative workspaces. This concept can revamp existing security solutions through the possibility of granting exceptions for moving objects entering danger zones. The downtimes created by incorrect intrusion detection will be reduced, and the RTLS technology can also monitor the flow of material in the process and optimize the trajectories of autonomous trolleys. Thus, the implementation of RTLS has multiple benefits for companies. The whole system can be modified according to the company’s needs. Another advantage is its compatibility with older robotic controllers. Its disadvantages are its lack of safety certification, the need to deploy RTLS (if not already available), and the longer per cycle computing time with higher camera resolutions.

The results of the experiments show how the camera resolution and number of objects in the workplace impact the image processing speed. The results show that lowering the resolution of the depth camera shortens the computation time. Keeping other things equal at the workplace, the computation time increased 9.60 times without HOG and 9.64 times with HOG when the depth camera resolution increased from 672 × 376 to 1920 × 1080. The noise rose from 3% to 10% for the mentioned resolutions. A more extended algorithm computation time means an increase in the safety system’s total reaction time and thus a necessary enlargement of the safety zones.

For the selected workplace and using the lowest camera resolution of 672 × 376, the radius of the danger zone size is 3200 mm, according to Equation (Equation 3). For a 1920 × 1080 resolution, the danger zone’s radius is increased to 5500 mm due to the longer processing time of the depth record.

In summary, we proposed an interesting and viable HRC concept with the application of alternative technologies to improve robotic workplaces’ safety and efficiency. Using a depth camera, safety zones were created around the robotic workplace and objects moving around the workplace were concurrently identified. The depth camera was also used to verify whether an identified object at the workplace is a person by using a Histogram of Oriented Gradients. The unequivocal identification of supply trolleys was achieved with the help of a real-time location system based on ultrawideband technology. The robotic manipulator’s speed of movement was regulated based on the objects identified and their distance from the robotic workplace. The benefit of this method lies in its ability to grant exceptions for supply trolleys that do not disrupt or jeopardize the operation of the robotic workplace, thus avoiding undesirable downtime. Based on our practical experience, we have not seen in industrial applications the interconnection of RTLS technology and depth cameras for the purpose of increasing the safety of robotic workplaces, as mentioned in the article, which we consider to be our main contribution.

Future work could focus on transferring the demanding image processing from a Central Processing Unit (CPU) to a Graphics Processing Unit (GPU), which could significantly reduce the computation time and provide an opportunity to increase the resolution of the processed image. One interesting alternative would be to use a CPU and GPU for simultaneous computation. The depth camera API offers the option to save a captured point cloud in the GPU memory. This would enable us to utilize the GPU’s computational power for some of the computational work, and the result would be sent to the CPU for further evaluation. It would be worth considering using the NVIDIA Jetson platform for image processing and evaluation or using cloud-based computing options for decentralized computations.

## Figures and Tables

**Figure 1 sensors-21-02419-f001:**
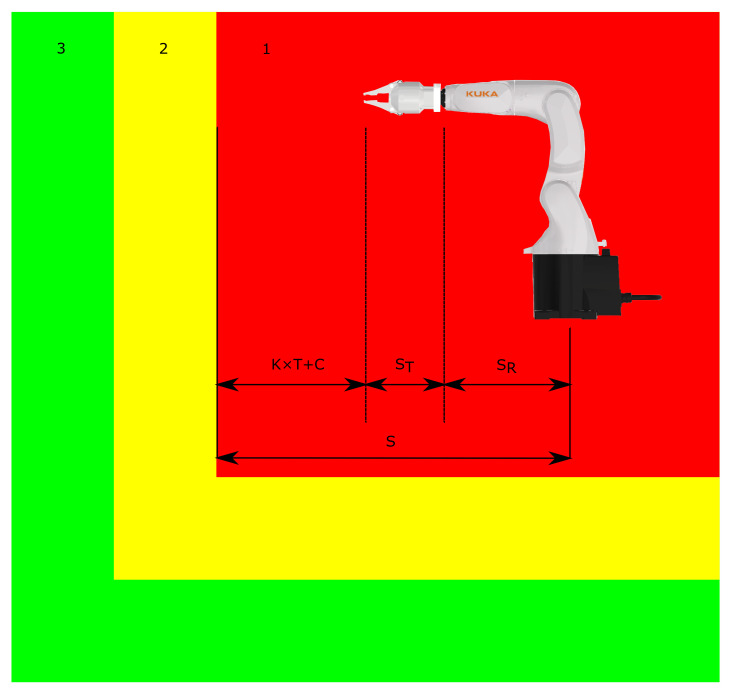
Minimum distance calculation example.

**Figure 2 sensors-21-02419-f002:**
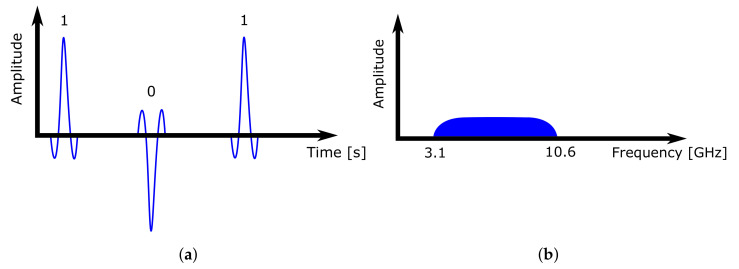
UWB communication pulses. (**a**) Time–domain behaviour. (**b**) Frequency–domain behaviour.

**Figure 3 sensors-21-02419-f003:**
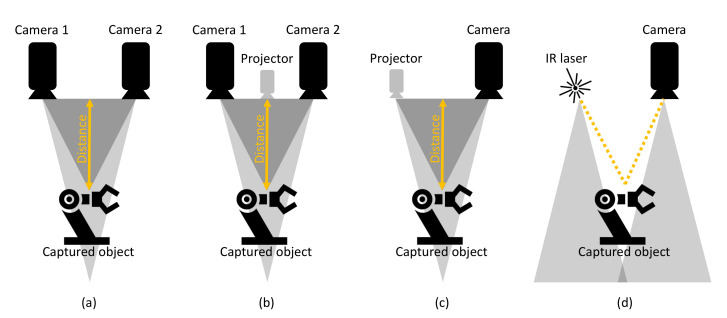
Depth camera variations. (**a**) Passive stereo vision. (**b**) Active stereo vision. (**c**) Structured light. (**d**) Time Of Flight.

**Figure 4 sensors-21-02419-f004:**
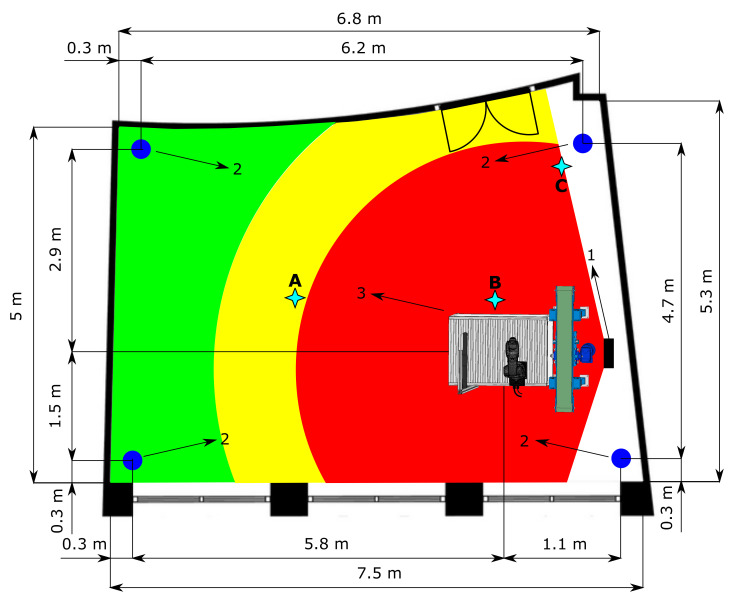
Test workplace floor plan. The red colour represents the danger zone, the yellow colour represents the warning zone, and the green colour represents the safe zone. (**1**) Depth camera. (**2**) RTLS anchors. (**3**) Robotic workplace. (**A**–**C**) RTLS measurement point, Table 3.

**Figure 5 sensors-21-02419-f005:**
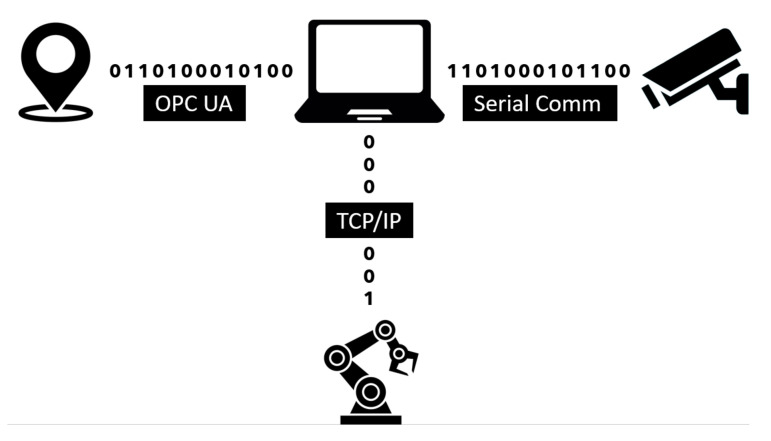
Architecture of the test system.

**Figure 6 sensors-21-02419-f006:**
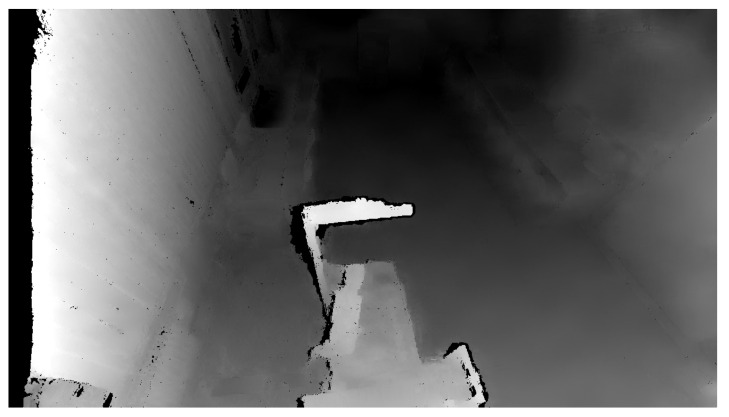
Depth image of the room.

**Figure 7 sensors-21-02419-f007:**
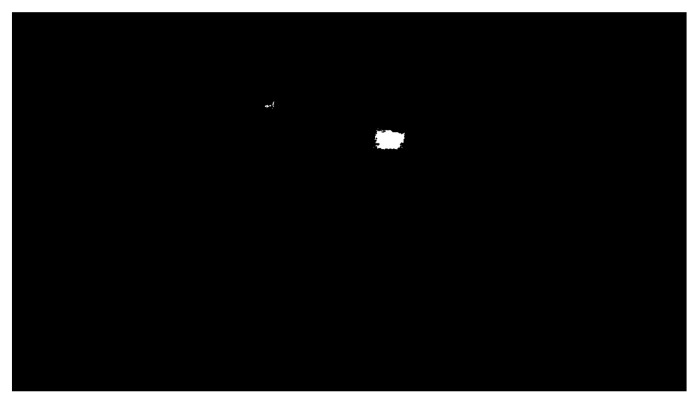
Highlight of a moving object.

**Figure 8 sensors-21-02419-f008:**
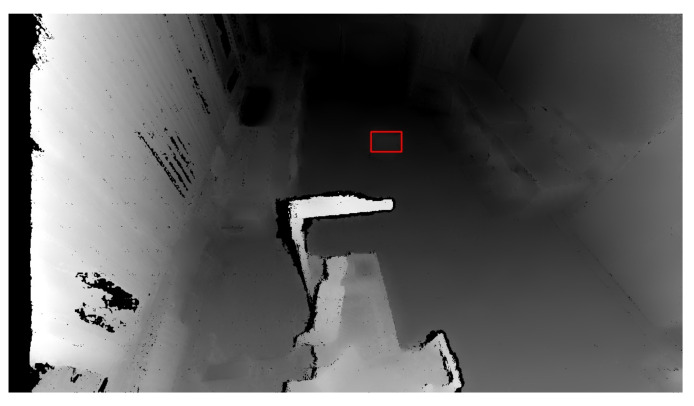
Marked and identified object on the depth image. The red rectangle represents an identified object based on depth change.

**Figure 9 sensors-21-02419-f009:**
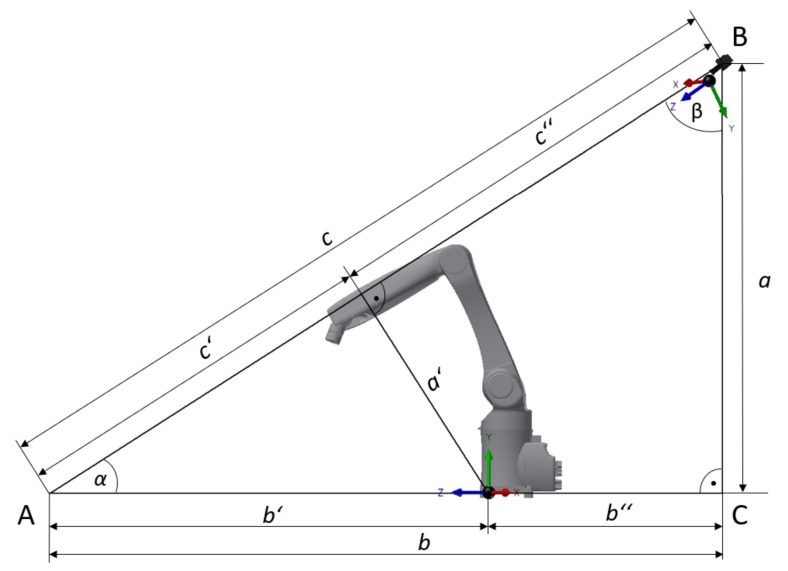
Side view of the workplace for the relative positions of the robot and camera coordinate systems.

**Figure 10 sensors-21-02419-f010:**
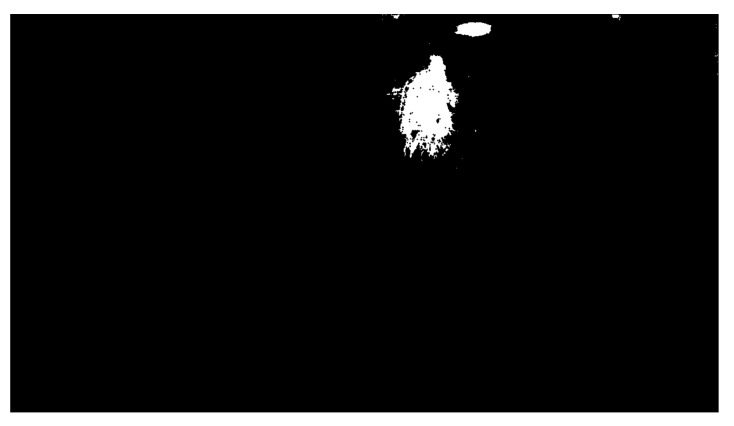
Object identification.

**Figure 11 sensors-21-02419-f011:**
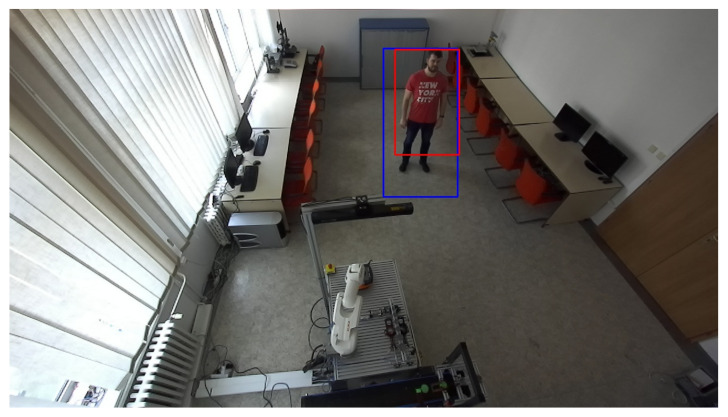
Verification of human presence. The red rectangle represents an identified object based on depth change. The blue rectangle represents an identified potential person based on the HOG algorithm.

**Figure 12 sensors-21-02419-f012:**
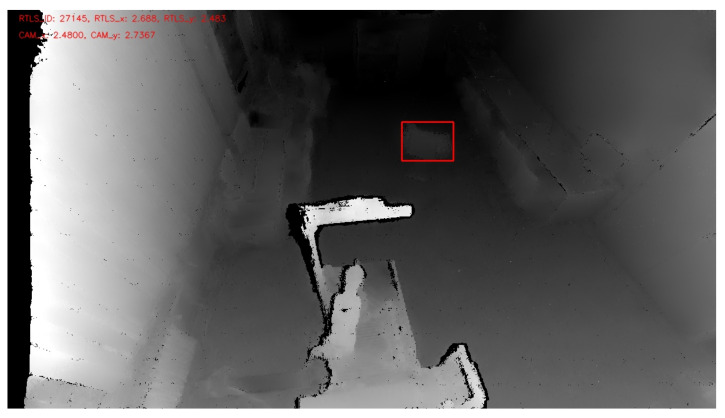
Supply trolley recognition with the help of RTLS and a depth camera. The red rectangle represents an identified object based on depth change, and its location is verified against the RTLS system.

**Figure 13 sensors-21-02419-f013:**
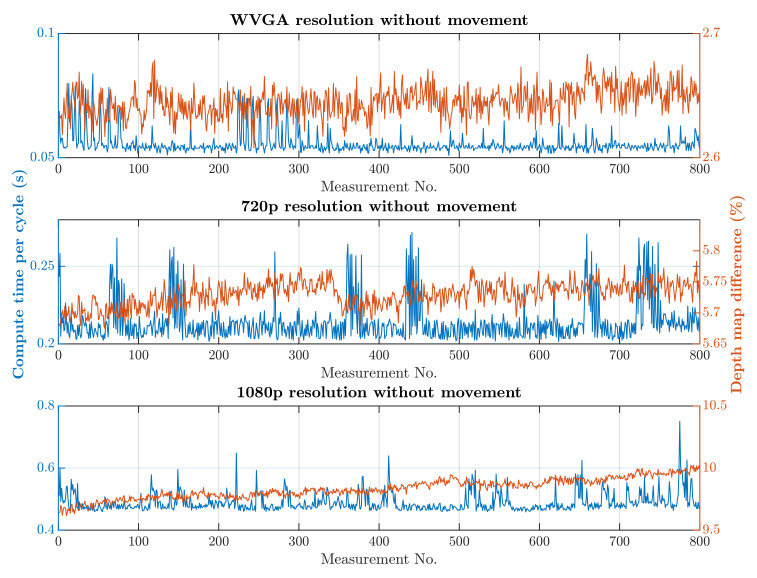
Results without HOG and without movement.

**Figure 14 sensors-21-02419-f014:**
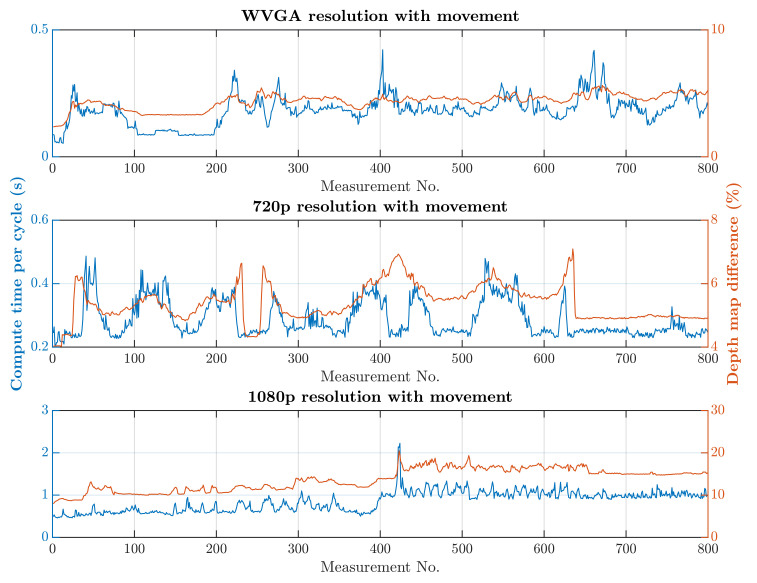
Results without HOG and with movement.

**Figure 15 sensors-21-02419-f015:**
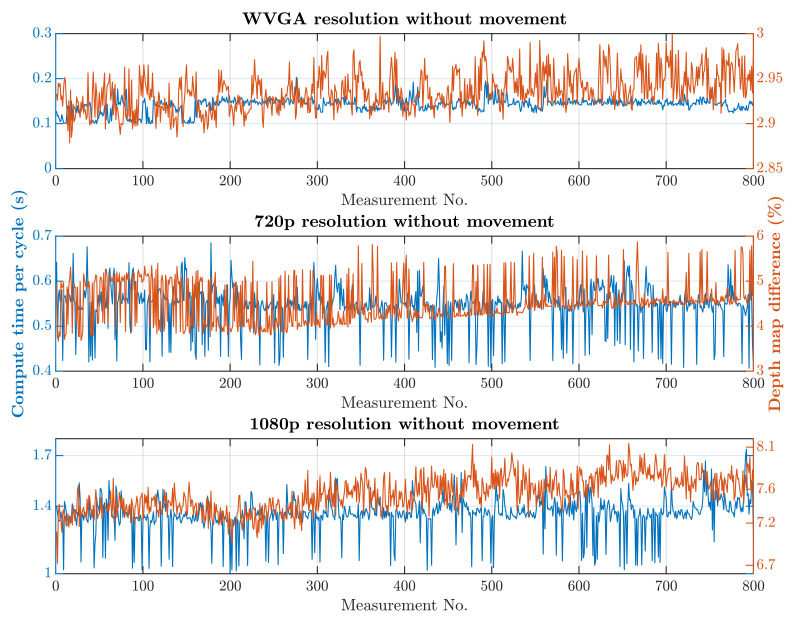
Results with HOG and without movement.

**Figure 16 sensors-21-02419-f016:**
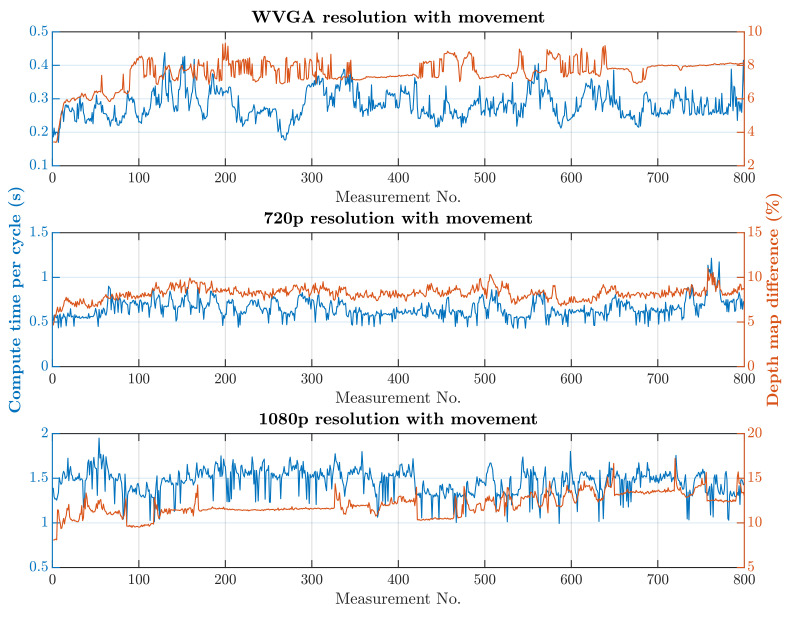
Results with HOG and with movement.

**Figure 17 sensors-21-02419-f017:**
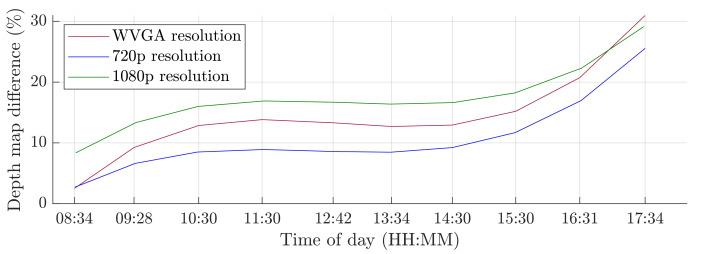
Influence of light changes during the day.

**Figure 18 sensors-21-02419-f018:**
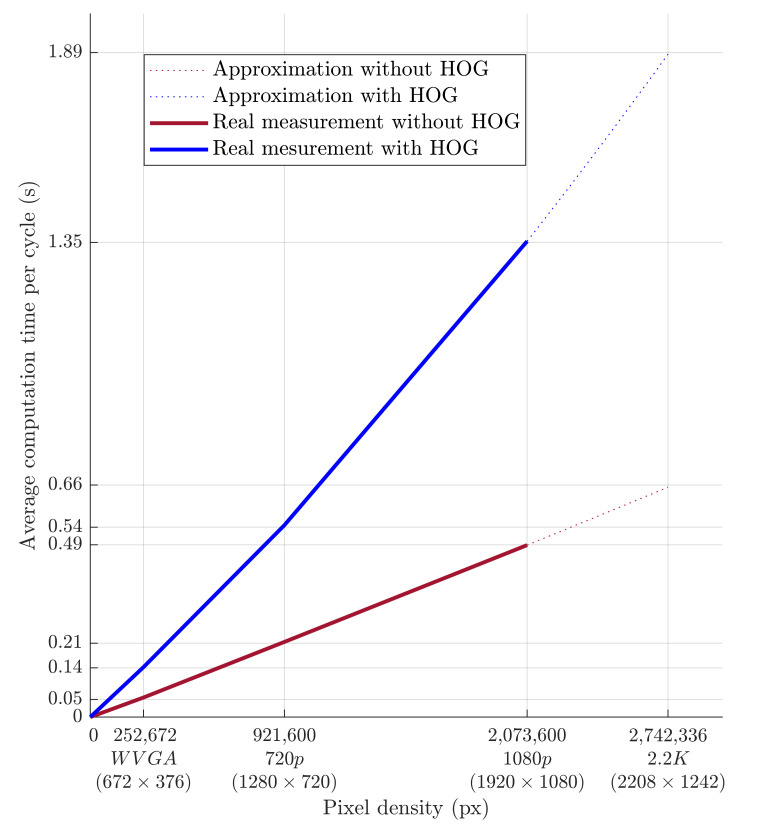
Dependence of the average computational time per cycle on the pixel count.

**Figure 19 sensors-21-02419-f019:**
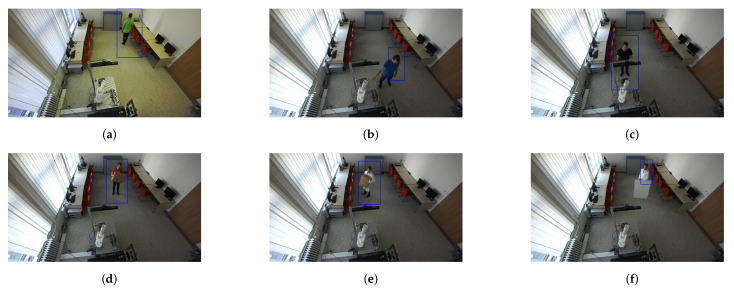
Validation of human recognition for various cases. The blue rectangle represents a potential person identified using the HOG algorithm. (**a**) A person wearing a green shirt. (**b**) A person wearing a blue shirt. (**c**) A person wearing a black shirt who is partially visible. (**d**) A person wearing a red shirt holding a small box. (**e**) A person wearing a white coat and holding a big box. (**f**) A person wearing a white coat behind a whiteboard.

**Table 1 sensors-21-02419-t001:** Speed limits for transient contact according to ISO/TS 15,066, which specifies safety requirements for collaborative industrial robot systems and the work environment.

	Tool Weight (kg)					
Body Part	1	2	5	10	15	20
	Maximum Robot Speed (mm/s)					
Palm/fingers	2400	2200	2000	2000	2000	1900
Upper hand	2200	1800	1500	1400	1400	1300
Lower hand	2400	1900	1500	1400	1300	1300
Stomach	2900	2100	1400	1000	870	780
Pelvis	2700	1900	1300	930	800	720
Upper leg	2000	1400	920	670	560	500
Lower leg	1700	1200	800	580	490	440
Shoulders	1700	1200	790	590	500	450
Chest	1500	1100	700	520	440	400

**Table 2 sensors-21-02419-t002:** Summary of collaborative industrial robotic systems properties.

Collaborative Type	Benefit	Power and Force Limited Robot	Traditional Robot
Safety-rated monitored stop	Promptly respond (stopping or moving)	Yes, but not always	Yes, requires additional sensors
Hand-guided	High variability of programs, quick changes	Yes	Yes, requires additional sensors
Speed and separation monitoring	Immediate resumption of higher speeds	Yes, but not always	Yes, requires additional sensors
Power and force limiting	Aplications requiring frequent operator presence	Yes	No

**Table 3 sensors-21-02419-t003:** RTLS accuracy based on the tag position within the test workplace.

UWB Settings	Tag Position Figure 4	Dispersion of Measured Values (m)	Update Rate (Hz)	Failed/Lost Measurements (%)
	A	0.17	3.07	21
Bitrate = 110 kbit/s Preamble length = 2048 symbols	B	0.25	3.00	21
	C	0.38	2.97	21
	A	0.14	43.47	0
Bitrate = 850 kbit/s Preamble length = 512 symbols	B	0.18	38.46	0.6
	C	0.20	35.71	0.3
	A	0.15	45.45	0.6
Bitrate = 6810 kbit/s Preamble length = 64 symbols	B	0.20	47.61	0.4
	C	0.21	58.82	0

## Data Availability

Not applicable.

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
