# Peer review of "Vision and RTLS Safety Implementation in an Experimental Human—Robot Collaboration Scenario"

_sensors, 2021, doi:10.3390/s21072419_

Round 1
Reviewer 1 Report
The paper deals with the safety of human-robot collaboration (HRC). The Authors well characterize the HRC safety standards and approaches to meet them, focusing next on the 'Speed and separation monitoring' (SSM) solution. This approach is considered in the context of a workplace consisting of an assembly robot and supply trolleys, as well as a Real-Time Location System (RTLS) and an RGB-D camera used for detecting and localization of objects appearing in the safety zones determined for this robotic cell. Both the RTLS and the vision-based detection concepts are clearly presented.
The main contribution of this work is a computer-based security system that using the information from RTLS and the camera supervises the robot speed. The paper considers two cases, with and without the presence of a human in the vicinity of the robot, provides the respective image processing algorithms and demonstrates their effectiveness through experimental results.
The described technology is sound and innovative in terms of the original adaptation of known theory and available software tools to the specific purposes of the constructed system. The considered solution is also well-motivated. Collaborative robots are more expensive and slower than classic industrial robots, whereas the proposed extension of the latter yields a safe and high-speed operating (on average) robotic system.
While overall the presentation of the material is good, there are a few issues that require clarification and/or correction.
1. The algorithms are incomplete. As output, robotic system speed is declared, yet no line defines how it is determined and what value it takes.
2. It is not clear - is the size of the safety zones invariant during the operation of the robot or does it depend on the position of the manipulator?
3. The rationale of the first scenario is not clear. If the only objects that can enter the zones are trolleys, why is it necessary to recognize them with the camera system instead of just capitalizing on the RTLS information about their position and ID?
4. Lines 247-251. The Authors might like to distinguish between robot control (the first process) and system supervisory control (the second process) that controls the operation modes of the robot based on the information from external sensors.
5. Lines 257-262. There is no distinction between a file that stores data and a process that uses it. In the paper, both are called SF.
Reviewer 2 Report
The paper describes an approach for endowing safety to a robotic workcell. The approach is based on the use of vision sensing for monitoring the environment, although localization motes have been also added to the trolleys that can enter in the restricted area around the robot. In general, the paper is well-structured and written, being ease to understand the problem and how safety can now be endowed in these workcells thanks to vision.
However, the authors must provide an deeper analysis of the current state-of-art. The use of vision for designing safe workcells is not new, and I encourage the authors to visit, for instance, the webpage of PILZ
Safe camera system SafetyEYE - Pilz INT
I am sure that the system described in this paper is able to provide additional features with respect to SafetyEYE, but I is important that this analysis must be added to the paper. Other alternatives such as FreeMove from Veo Robotics should be also include in this analysis.
On the other hand, the validation of the system should be also improved. It is important to demonstrate if the system delays will allow to quickly detect the presence of a person close to the robot. The ability of the HOG approach for detecting people in motion should be emphasized in the results analysis. It is also important to show how the possible changes on the illumination conditions can alter the normal execution of the system, as it could be also important to evaluate the restrictions to this normal and correct execution. A robustness evaluation is needed: is the system able to detect, in all cases, the human presence? what does it happens when only a partial view of the person is available? or when the person is carrying a package, is this person detected by the system? More examples showing these situations and the robust response of the system must be considered. They are not rare situations in this scenario.
Reviewer 3 Report
This paper presents the important topic of safety during the human robot interaction during the collaborative task. The authors presented the essential background information for the safety framework and the standards associated with industrial robots and safety. Subsequently, the authors demonstrated the use of depth-based camera and Ultra Wide Band (UWB) technology for real time location of the objects in the test workplace floor. The paper uses standard libraries and methods for implementing the necessary safety feature. The paper is well written in general however following points are listed for authors:
Avoid abbreviations (DOF and OAT) in Abstract. Abbreviations should be introduced at the first place where they appear in the main text.
In the Introduction section, the author can highlight the use of monocular camera as a sensor to estimate the 3D position data. The work-related to monocular camera and the quality of measurement should be cited like “A geometric approach for kinematic identification of an industrial robot using a monocular camera”, “A self-calibrating probabilistic framework for 3d environment perception using monocular vision”, etc.
No need to place comma and fullstop after the equations in (1), (2), etc.
In the Background, section authors are suggested to provide the robot's details, with its DH parameters and speed limits of the robot.
Figure 4 should be supported with the dimension of the test floor settings.
(Line 263-270) : The make of the RGBD camera can be mentioned. Also, the rationale behind the placement of the camera and UWB sensor. Is it the optimal position, or is it that the camera can be placed anywhere inside the workplace.
Does the proposed work will get affected by the lighting conditions in the workplace.
The UWB interference is observed when it is kept close to the metal surface. Authors should provide the real experimental setup images with the placement of UWB, camera, and tracker placed on the trolley. Comment on the error observed due to the location of the UWB is encouraged.
In the result section, the comment should be made on the movement of the trolley or robot along with its speed.
The results should highlight the measurement accuracy of the system at various speeds.
The framework for interaction is only presented for a single human and trolley. What will be the situation in the case of two or more trolleys and two or more humans? What is the consideration of safety interaction between trolley and human?
Grammar and spelling check is suggested.
Reviewer 4 Report
This article focuses on safety in Human-Robot Collaboration workplaces and presents an implementation based on computer vision and Real-Time Location Systems. Overall, the paper presents serious flaws, and the research study needs to be reformulated. The main drawbacks and minor suggestions are described below:
Main drawbacks:
- The contribution and novelty are minimal. It is not clear neither what the contribution is: is it the combination of existing technologies? Where is the novelty? The authors say: "The aim of this article is to verify the application use of depth cameras to make robotic workplaces safe with zoning." Is this a new solution?
- There is no comparison with existing technologies. Why is this "solution" better than existing solutions.
- In Section 2, line 139, the authors mentioned: "When choosing an industrial robot, the price needs to be considered in addition to the type of task. Collaborative robots are more expensive and slower than classic industrial robots. It is, therefore, more advantageous to keep the classic industrial robot but supplement it with safety and collaborative elements. In a workplace where human interaction with a robot is expected, the most appropriate collaboration methods are safety monitored stop or speed and separation monitoring." What about the price of the proposed solution? The price of an RTLS system is also very high. Then, why is it better an expensive RTLS-based solution than using cobots?
- The major issue is that the method/contribution is not explained. After the 'huge' Background Section, which does not introduce any novelty, the reader goes directly to the results. Between these two sections, one expects an explanation of the proposed solution or approach; otherwise, it seems that all that was done in this work is the integration of existing technologies for a problem that already has multiples solutions (including ISO norms).
- The manuscript must be reformulated; the structure needs to be improved. E.g., First, introduce the motivation, problem tackled, existing solutions, and their limitations; second, explain your approach/contribution (not in the paper). Then, define the experimental evaluation (in a real-world scenario if possible) and present the results. After that, discuss your results, compare them with existing solutions (not in the paper), and explain the benefits and limitations of your contribution (not in the paper neither). Finally, present the conclusions of the work supported by the results.
Minor comments/suggestions
- Present the structure of the document at the end of the introduction.
- In Fig. 2, what do the colors mean?
- What about the use of event cameras for the detection of moving objects?
- The paper can be summarized: Section II and Fig. 7, 8, and 10 present information that one can easily find in literature.
- In Fig. 11, what do the blue and red rectangles mean?
- What are the outcomes of this research? Include a brief description at the end of the abstract.
Round 2
Reviewer 2 Report
The authors have included in the revised version of the paper all suggestions provided by reviewers. Comparison with other proposals have been considered and experimental evaluation is now more complete. I consider the paper is ready for publication.
Author Response
Thank you for the review and your time.
Reviewer 3 Report
The paper has incorporated the comments suggested by the reviewers.
The authors are suggested to go through the language and grammar check once again, still there are few errors present which must be improved.
Author Response
Thank you for your time and the review. English got now improved by the MDPI English Editing Service.Reviewer 4 Report
My comments from the first round of reviews have been addressed. Now the manuscript presents some improvements w.r.t. the previous version:
- First, the structure of the paper has improved, in my opinion. Now the introduction and the methodology are presented more clearly. The integration of section 3 helps the reader, improving the readability of the manuscript.
- The introduction and preliminaries sections have also been improved.
- The contribution and novelty are now clearly stated in the manuscript.
- Besides, some minor changes have been considered in several parts of the document.
All these aspects are related to the quality of the manuscript, which, in my opinion, has been improved significantly. In this regards, I would like to thank the authors for considering and addressing all my comments.
However, regarding the contribution itself, for me, it is still not novel enough for a journal paper. Even though the authors claim that the novelty of the work lies in "the integration of a real-time location system into a vision-based safety system of our own creation. Which is a new solution in the field of HRC"; this is, in my opinion, not a very accurate statement. It might be true that the integration of both technologies is something that was not done before, but still, both technologies already exist. Hence, there is no novel contribution from the scientific point of view, which is, in my opinion, the most fundamental aspect of a scientific journal paper.
In any case, this is my personal opinion, and if the rest of the reviewers and the editor consider that the contribution and novelty are enough for its publication in the journal, I would not argue more in this respect, in which I think I clearly stated my position. Despite the main drawback aforementioned, all other aspects of the paper are correctly addressed.
Author Response
Thank you for your time and the review. English got improved by the MDPI English Editing Service.
Yes, we fully agree that these technologies are not new. Based on our experience in industrial companies with which we actively cooperate, we see our contribution in the interconnection of vision and RLTS systems, which we have not yet seen.
It is about the synergy of the interconnection of two co-existing technologies intended in principle for different deployments to increase the safety of robotic workplaces, as mentioned in the article.
We think that the novelty doesn't need to be always designing new technologies. If an application of certain known technologies is made for the first time, we think this should also be classified as a novelty.